# Experimental Investigation on the Performance of SBS-Modified Asphalt Waterproofing Membrane by Thermo-Oxidative Aging and Freeze–Thaw Cycle

**DOI:** 10.3390/polym16182593

**Published:** 2024-09-13

**Authors:** Juanjuan Wang, Xian Li, Guojun Sun, Xingpeng Ma, Hongwei Du

**Affiliations:** 1China Electronics Engineering Design Institute Co., Ltd., Beijing 100142, China; 2China Electricity Council, Beijing 100761, China; 3Faculty of Architecture, Civil and Transportation Engineering, Beijing University of Technology, Beijing 100124, China; maxingpeng@emails.bjut.edu.cn (X.M.);

**Keywords:** SBS-modified asphalt waterproof membrane, thermal-oxidative aging, freeze–thaw cycle, low-temperature cracking performance

## Abstract

With global warming in recent years, extreme weather conditions have increased in frequency and intensity, exacerbating the challenges for waterproofing materials. The current stages of SBS asphalt waterproofing membrane aging research mainly focus on the raw materials and modifiers in a single factor; multifactor-coupled aging research is less studied. This study focused on the coupled aging characteristics of SBS-modified asphalt waterproofing membranes, aiming to reveal the mechanism of its influence on the material’s performance under the environmental effects of high temperature and freeze–thaw. Through the accelerated aging test, we simulated the environmental conditions of high temperature in summer and freeze–thaw in winter to observe the mechanical properties of waterproofing membranes, low-temperature flexibility, and apparent phenomena. Then, Fourier transform infrared spectroscopy (FTIR) evaluated the performance and chemical structure of SBS-modified asphalt waterproofing membranes after aging by the coupled aging of the thermo-oxidative freeze–thaw cycle. The results showed that the low-temperature flexibility of the waterproofing membranes was significantly reduced after the coupled aging effect, and, at the same time, their tensile strength was also reduced. However, despite the tensile properties being impaired, the membrane maintained good ductility, and its elongation at break did not fall below 47%.

## 1. Introduction

Asphalt waterproofing membranes are widely used in civil engineering for waterproofing and are the most commonly used material in building roofs and underground storage and in performing bridge deck renovation [1], as shown in Figure 1, playing a vital role in keeping these spaces dry.

Bitumen is the main ingredient in the preparation of waterproofing membranes and is a complex hydrocarbon, but it is prone to flow at high temperatures and to embrittlement at low temperatures, limiting its further application. Aging phenomena are highly susceptible in production and use due to heat, UV radiation, and atmospheric agents, which are some of the most common phenomena that affect the mechanical, physical, and chemical properties of binders [2]. So, a variety of modified substances are added to the asphalt membrane. SBS-modified asphalt waterproofing membrane is manufactured by coating a modified asphalt-coating material (composed of asphalt, modifier, fillers, etc.) onto the surface of a substrate, then smoothing it with a roller and finally applying polyethylene film to both sides of the coating material. Due to its excellent resistance to high-temperature sagging and low-temperature cracking, it holds a dominant position in the Chinese waterproofing-membrane market.

The SBS-modified asphalt waterproofing membranes laid on the roof, due to long-term direct exposure to sunlight and the erosion of the natural environment, will become deformed or begin peeling and thus cause cracking of the concrete surface of the roof. Meanwhile, due to the effects of the long-term temperature differences between winter and summer, the water seepage phenomenon causes the waterproofing membrane’s surface to crack. This relates to the aging resistance of the waterproofing membrane itself, so the study of the aging mechanism of the asphalt membrane is essential to ensure the quality of roof waterproofing.

In recent years, many scholars have researched topics related to the aging of asphalt materials and SBS-modified asphalt membranes. Singh et al. [2] simulated the short-term and long-term aging of base material and asphalt binder by using the Rolling Thin-Film Oven Test (RTFOT) and the Pressure Aging Vessel Test (PAV), respectively; investigated the properties of asphalt aging by scanning electron microscopy SEM and Fourier transform infrared (FTIR) spectroscopy; and explained the reasons for aging. Mazumder et al. [3] summarized the possible reasons for the degradation of asphalt morphology and properties after aging by using scanning electron microscopy and concluded that the binder has a certain resistance to cracking. He et al. [4] investigated the effects of thermal aging and UV-thermal synergism on the properties and chemical structures of high- and low-doped SBS-modified asphalt waterproofing membranes and found that the changes in the softening point and the low-temperature flexibility after thermal aging were much smaller in the high-doped waterproofing membrane than in the low-doped. At the same time, fluorescence microscopy was used to observe that the high doping showed a network structure which was only partially destroyed after thermal aging, and the structure was destroyed after UV-thermal synergistic aging, while the low doping showed an island structure, which almost disappeared after aging. Garrido et al. [5] conducted peel tests on different connections of old and new waterproofing membranes, taking into account the aging time and the type of membrane, and concluded that APP (polypropylene-modified asphalt)’s tensile properties are superior to those of ordinary SBS-modified asphalt membranes. Marques et al. [6] tested water and thermal aging for roof waterproofing membranes and investigated the effect of aging on the adhesion of the self-protecting granules on their surfaces. The aging of asphalt is the main reason for its service life’s shortening [7], and some scholars have investigated the aging properties of SBS by altering the admixtures in the prepared asphalt. Di et al. [8] added graphene oxide to ordinary asphalt waterproofing membranes and found that the addition of graphene oxide led to a significant improvement in its chemical properties at high temperatures and under UV, as well as improvement in its waterproofing properties. The aging of asphalt is a complex process; Feng et al. [9] investigated the changes in the properties of asphalt with three anti-aging agents, and a new type of waterproofing membrane was made by mixing asphalt with waste tire particles [10,11], which also improved its mechanical properties. He et al. [12] added organic montmorillonite (OMMT) to the study of the aging resistance of SBS-modified waterproofing membranes. The results showed that it can improve its anti-aging properties.

In addition, the low-temperature flexibility of waterproofing membranes is also an important indicator, which is the flexibility of waterproofing membranes in the specified low-temperature temperature after experiencing a certain period of and being subjected to bending. Low-temperature flexibility is a direct reflection of how well the waterproofing membrane in the low-temperature environment is able to maintain the ability to elongate or maintain resistance to grassroot cracking. If the low-temperature flexibility of the waterproofing membrane is poor, the materials harden and crack easily in cold conditions, making the waterproofing effect much lower. The thermal and oxygen stability of the raw materials is an important reason for the low-temperature flexibility of modified asphalt waterproofing membranes; Chen [13] and Cui [14] carried out low-temperature long-term and high-temperature short-term aging of raw materials, respectively, and the tested low-temperature flexibility for the prepared asphalt rolls. These tests found that the low-temperature flexibility of the rolls was significantly reduced after high temperatures. In addition, Ge et al. [15,16] investigated the low-temperature properties of the membranes after heat aging and water-immersion aging and found that water-immersion aging had a large effect on their low-temperature flexibility. Huang et al. [17] conducted water freeze–thaw and salt freeze–thaw cycling tests on rubber-modified asphalt to describe the evolution of the low-temperature properties under freeze–thaw cycling conditions at the macroscopic level and found that freeze–thaw cycling reduces the low-temperature flexibility of the asphalt binding material. There are various factors affecting the performance of roofing-asphalt waterproofing membranes; up until now, the performance study of waterproofing membranes mostly focused on the effect of single factors such as high temperature, low temperature, light [18], etc., and fewer studies consider the degradation of the performance of the multifactorial (natural environment) [19,20], and the cost of the test is high. Therefore, it is necessary to carry out performance studies on multifactor-coupled aging.

In this paper, we take the roofing SBS-modified bitumen waterproofing membrane as the research object and consider the influence of the environment on its performance. When the low-temperature flexibility was used as an indicator to assess the degradation of performance under the action of thermo-oxidation and freezing-thawing coupling, SEM and FTIR revealed its degradation mechanism from the microscopic. Finally, we carried out tensile tests on the coupled aging specimens and summarized the impact on the mechanical properties of SBS asphalt waterproofing membranes, which guided the actual engineering.

## 2. Experimental

### 2.1. Materials

All test materials were sourced from the same batch produced by Oriental Yuhong Enterprise. The model of the roll is SBS I PY PE PE 3, where SBS refers to the use of modified bitumen with styrene-butadiene-styrene block copolymer. I denotes that it is a Type I waterproofing roll-roofing. PY stands for polyester substrate, indicating that the base material of the roll is polyester fabric. PE represents the polyethylene membrane used as the surface coating material. The number 3 specifies that the thickness of the roll is 3 mm. Thus, SBS I PY PE PE 3 describes a 3 mm-thick SBS-modified bitumen waterproofing roll-roofing material with a polyester substrate and a polyethylene surface coating. The material is shown in Figure 2, and its main performance indicators are shown in Table 1.

### 2.2. Test Program

To simulate the performance degradation of the waterproofing membrane under high-temperature and freeze–thaw conditions, a thermo-oxidative aging and freeze–thaw cycle test was conducted on the SBS-modified asphalt waterproofing membrane. The thermo-oxidative aging test considered the highest temperatures that roofs and external walls can reach during summer [21], which was set at 80 °C with aging time as a variable. The aging specimen was placed in a constant temperature and humidity-curing box for 24 h for a freeze–thaw test. The freeze–thaw cycle in the automatic freeze–thaw cycle box was programmed for a −20 °C environment maintenance for 12 h and then was programmed for a 23 (±2) °C constant temperature box maintenance for 12 h; this was one cycle. After the completion of coupled aging following the requirements of GB18242-2008 “Styrene butadiene styrene (SBS) modified bituminous sheet materials” [22], low-temperature flexibility and tensile performance testing were carried out on the specimens, and the tensile speed was 100 mm/min. For the low-temperature flexibility tests, test at least five specimens for each coupled aging time were tested, and it was ensured that at least four samples cracked in each set of tests. For tensile tests, we used three specimens per set. Finally, the mechanism of degradation of the SBS waterproofing membrane aging performance was further explained by FTIR. The test program is shown in Table 2 below.

### 2.3. Specimen Preparation and Test Equipment

For the coupled aging test, the specimen was cut into a rectangle measuring 150 mm × 25 mm, and the specimen was cut uniformly from the direction of the width of the coil with the long edge along the longitudinal direction of the coil, and the cut was not less than 150 mm away from the edge of the coil. The tensile specimen size was 300 mm × 50 mm, as shown in Figure 3 below. The specimens were uniformly laid flat in a thin-film oven (TFO) with a freeze–thaw cycle tester (FTCT) to simulate high temperature and a freeze–thaw environment and were tested for low-temperature flexibility on a low-tenderness apparatus. The specimens were also tested for tensile strength and elongation at break on a universal testing machine. The tests are shown in Figure 4. FTIR was used to characterize the changes in the chemical framework structure of the SBS-modified asphalt before and after aging, A small number of samples were taken and placed in the diamond ATR module that had a wave number range of 4000~400 cm^−1^, scanning times of 16 times/min, and a resolution of 4 cm^−1^.

## 3. Results and Discussion

### 3.1. Apparent Appearance and SEM Analysis

After thermo-oxidative aging, the coil-cladding coating gradually hardened, and the rectangular specimen underwent minor deformation. The volatilization of light components in the asphalt led to the release of an odor, with the surface emitting a distinct smell of heated asphalt. This is a typical feature of the sub-oxidative degradation of asphalt rolls [23]. The surfaces of specimens with different aging durations are shown in Figure 5. The surface of the unaged specimen was smooth, with a clear texture and glossy appearance, as shown in Figure 5a. As the aging time increased, the edges of the specimen’s surface began to bulge, and the surface turned a yellowish hue. This observation aligned with Zhou et al.’s findings on the broader dispersion of composite-modifier chromatographic peaks after aging [24]. A small number of wrinkles appeared on the surface of the modified asphalt after aging, which could be attributed to the differing thermal expansion coefficients of asphalt and polymers, resulting in an uneven contraction of the material surface.

Figure 6 shows the unaged and the coupled-aging, SBS-modified asphalt waterproofing membrane electron microscope scanning image. From the microscopic view, the surface of the unaged specimen is smooth, flat, and dense. As the aging progresses, the surface gradually becomes rough and develops a layered, concave-convex structure. This corresponds to the macroscopic observation of small pits and cracks of varying degrees on the surface of the SBS-modified bitumen waterproofing membrane. It is hypothesized that the temperature stress generated during the freeze–thaw cycle causes surface damage to the asphalt binder. Simultaneously, the volume expansion and contraction during the freeze–thaw cycle lead to internal intermolecular interactions. When the temperature stress is insufficient to counteract the material’s stress-relaxation properties and exceeds the tolerance limit of the asphalt mixture, tiny cracks form [25]. Furthermore, as the number of freeze–thaw cycles increases, the resulting damage becomes more pronounced [26].

### 3.2. Quality Loss Rate

The quality of the SBS-modified asphalt waterproofing membrane samples with different aging times was tested, and the results are shown in Figure 7. From the figure, it can be seen that after different aging times, the quality has decreased, which is because, under heat, the light component in the bitumen volatilized from the surface of the material, resulting in a decrease in the quality of the waterproofing membrane. The rate of quality loss was faster in the first 5 days and slower in the 7th to the 28th day, but the overall trend was positively correlated with the aging time. When measuring the mass, the asphalt coil softened from heat, and some white powder adhered to its surface crevices, leading to a decrease in mass loss. This occurred during the thermal-oxidative aging test of the coil. To prevent the specimen from adhering to the internal platform of the film oven, putty powder was applied. As the aging time increased, the specimen’s surface became more adhesive, causing the putty powder to stick to it. The post-processing could not completely remove the powder, resulting in an apparent increase in the specimen’s mass, The specimen was specifically shown in Figure 8. Yang et al. also compared the thermal aging and mass loss of Type I SBS-modified asphalt and found that the rate of mass loss was positively correlated with the thermal aging time, though the curve did not follow a linear pattern [20]. The aging of the asphalt due to oxidation mechanisms resulted in a reduction in mass [27].

### 3.3. Low-Temperature Flexibility

A low-temperature flexibility test was performed on the waterproofing membrane specimens after thermo-oxygen aging and the freeze–thaw cycle; the results are shown in Figure 9. The low-temperature flexibility of the unaged specimen was −29.5 °C, which fully met the requirements of GB 18242-2008 “Styrene butadiene styrene(SBS) modified bituminous sheet materials.” The low-temperature flexibility of the SBS-modified asphalt waterproofing membranes continued to decrease as the coupled aging proceeded, eventually decreasing to −16 °C, which was even more severe for the specimens that experienced only thermal-aging temperature decay [20]. During the thermo-oxidative aging, the active groups in the asphalt underwent cleavage, producing more reactive free radicals. This phenomenon accelerated the reaction between the asphalt and oxygen, resulting in increased hardening and brittleness, which significantly reduced its low-temperature flexibility [28]. Simultaneously, the butadiene segments in the SBS also formed free radicals under thermo-oxidative conditions, leading to molecular chain breakage and degradation. As the molecular chains broke, the SBS gradually lost its rubbery elasticity, weakening its ability to enhance the material’s low-temperature performance [29]. Additionally, the butadiene chain segments in SBS were more prone to breakage during freeze–thaw cycling, which was the primary cause of the material’s degradation when subjected to both thermo-oxidative and freeze–thaw effects. A similar explanation is provided in the analysis of the infrared spectra in the following subsection.

### 3.4. FTIR Analysis

The changing pattern of infrared absorption values at different wavelengths has been widely used in the study of polymer materials in the degradation of properties. A quantitative link between the degree of aging and rheological indexes is now a well-established means of analysis by Fourier transform infrared (FTIR) analysis, as well as tracking the changes in oxygen-functional groups. Figure 10 shows the FTIR of the unaged and the coupled aged specimens, and the information obtained from them about the peaks is given in Table 3.

The spectrograms revealed peaks near 2920 cm^−1^, 2850 cm^−1^, 1460 cm^−1^, 1017 cm^−1^, and 720 cm^−1^ in the spectral test results of each specimen. The peaks near 2920 cm^−1^ and 2850 cm^−1^ were characteristic peaks due to symmetric and antisymmetric telescopic vibrations of the methylene groups, mainly originating from aromatic compounds in the bitumen. The characteristic peak near 1460 cm^−1^ was generated by the in-plane deformation vibration of the methylene groups. The characteristic peak at 1017 cm^−1^ was generated by the S=O stretching vibration of the sulfinyl group, and the characteristic peak at 720 cm^−1^ was generated by the N-H out-of-plane bending vibration.

When comparing the FTIR of the SBS-modified asphalt waterproofing membrane specimens before and after aging, it can be seen that the aging specimen at 875 cm^−1^ displayed a new characteristic peak, which may be due to the high-temperature freeze–thaw cycle aging that changed the C-H surface-bending vibration of the characteristics of the peak. With the increase in temperature and freeze–thaw cycle aging times, the -OH peak at 1460 cm^−1^ kept getting bigger, which was due to the oxygen uptake reaction of the SBS asphalt waterproofing membranes during thermo-oxidative aging in the test chamber. As can be seen from the figure, with the increase in the aging time, the peak value of the sulfenyl group at 1017 cm^−1^ was significantly larger than the peak value of the unaged SBS-modified asphalt waterproofing membrane specimen; this was due to the sulfur component of the asphalt in the coupling of the aging process, which occurred in the oxygen-absorbing aging. Because of this, when the SBS molecular chain segments in the C=C unsaturated double-bond dissociation energy decreased, it was easy to observe the degradation reaction in the hot air, and, at the same time, the sulfur in the asphalt reacted with oxygen to produce sulfinyl groups.

### 3.5. Tensile Properties of Waterproofing Membranes

The tensile properties of the unaged and coupled-aged SBS-modified asphalt waterproofing membranes were tested, and one set of results is shown in Figure 11. The overall trend of the tensile curves of the specimens with different aging times was consistent, and the initial stiffness was consistent, but the final breaking force was different, and the maximum tensile force did not monotonically increase or decrease; instead, it fluctuated. Additionally, most of the breaking force after the coupled aging was lower than that of the unaged specimens, and only the specimen with coupled aging for 28 days still maintained a large breaking force, which may be related to the inhomogeneity of this part of the specimen. Moreover, changes in the internal structure of the material during the aging process may have contributed to this behavior. The material likely experienced a dynamic equilibrium between cross-linking and degradation. At certain stages, cross-linking enhanced the material’s strength, whereas at other stages, degradation became dominant, resulting in a reduction in strength. Additionally, the gradual formation and propagation of microcracks may have further contributed to the fluctuations in the tensile force [30].The tensile damage morphologies of the coil specimens under different aging time lengths were similar; the stretched specimens’ cross-section was elongated and eventually damaged in the middle or end and the broken length was much larger than the length of the unstretched specimen and was unable to recover its deformation, as shown in Figure 12. Calculating the maximum elongation according to the following formula, it was found that the elongation was close to 50%, the results are shown in Figure 13 and, with the increase in the coupled aging time, the maximum force elongation also increased gradually; that is, the ductility of destruction was increased gradually.
Maximum elongation = deformation of the sample at maximum tension/clamping distance × 100%.

## 4. Conclusions

To investigate the influence of high-temperature freeze–thaw-coupled aging on the performance degradation of SBS-modified asphalt waterproofing membranes, this study designed and implemented a thermo-oxidative freeze–thaw cyclic test to simulate the accelerated aging effect of the membrane in a real environment. The following conclusions were made through the performance test of the specimens:

(1) With the increase in the coupled aging time, asphalt waterproofing membranes will decrease the low-temperature flexibility, and cracking at higher temperatures, compared to only thermo-oxidative aging environment, the low-temperature flexibility of the SBS-modified asphalt waterproofing membranes is worse. The reason for this is that the SBS molecular chain C = C is unsaturated and easily combined with oxygen under coupled aging. During freeze–thaw cycles, the temperature stress in the specimen leads to surface damage more easily.

(2) The quality of the SBS-modified asphalt waterproofing membranes decreased after the coupled aging test, and the weight loss in the first 5 days is obvious, which is related to the test conditions. The test conditions can be improved to provide experience for future related tests.

(3) With the increase of the coupled aging time, the SBS-modified asphalt waterproofing roll-roofing tensile breaking force changes are not significant, but its tensile elongation with aging time increases and becomes larger.

## Figures and Tables

**Figure 1 polymers-16-02593-f001:**
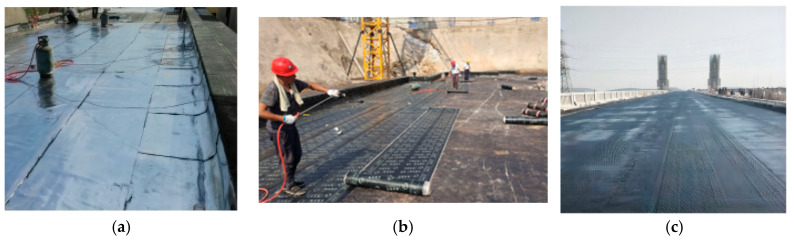
Waterproofing membrane application scenes: (**a**) building roof, (**b**) underground storage room, and (**c**) bridge decks.

**Figure 2 polymers-16-02593-f002:**
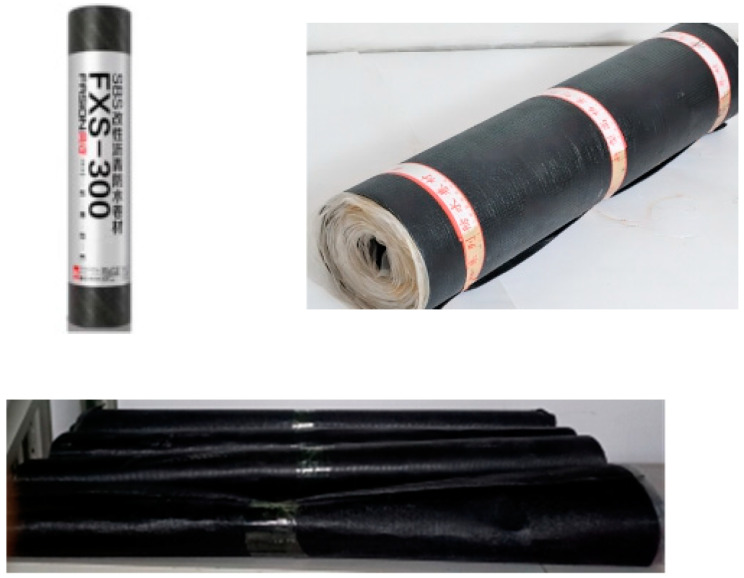
SBS-modified asphalt waterproofing membrane.

**Figure 3 polymers-16-02593-f003:**
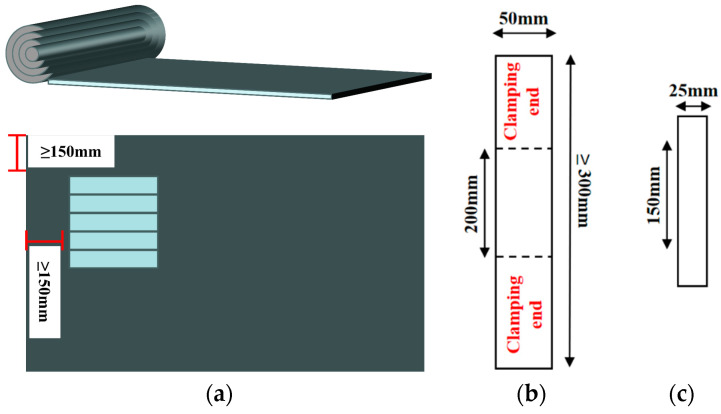
Specimen preparation and dimensions: (**a**) specimen cutting direction, (**b**) tensile specimen dimensions, and (**c**) coupled aging specimen dimensions.

**Figure 4 polymers-16-02593-f004:**
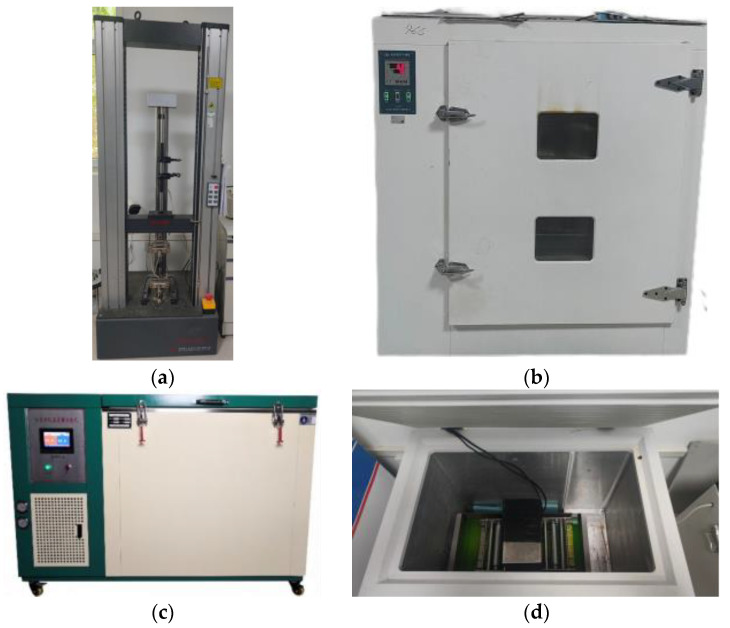
Test equipment: (**a**) universal testing machine, (**b**) thin-film oven, (**c**) freeze–thaw cycle test cabinet, (**d**) low-tenderness apparatus and (**e**) SEM; (**f**) Fourier transform infrared (FTIR) Spectroscopy.

**Figure 5 polymers-16-02593-f005:**
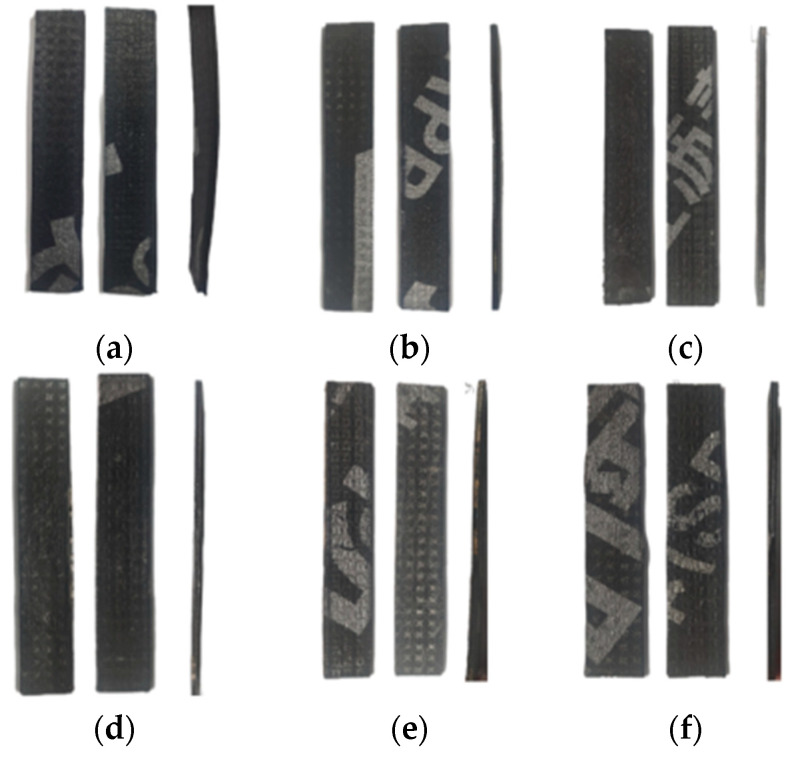
Appearance of specimens: (**a**) unaged, (**b**) aged 2 days, (**c**) aged 5 days, (**d**) aged 7 days, (**e**) aged 14 days, and (**f**) aged 28 days.

**Figure 6 polymers-16-02593-f006:**
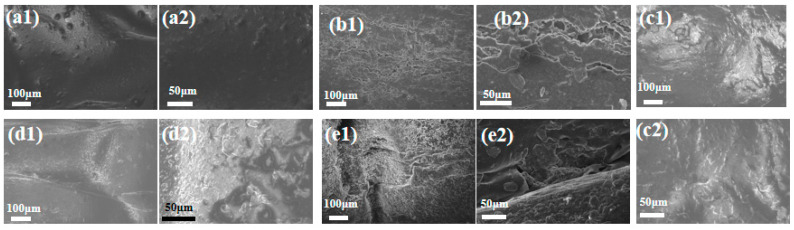
Scanning electron micrographs before and after aging: (**a1**) 100 μm, unaged, (**a2**) 50 μm, unaged, (**b1**) 100 μm aged 2 days, (**b2**) 50 μm aged 2 days, (**c1**) 100 μm aged 5 days, (**c2**) 50 μm aged 5 days, (**d1**) 100 μm aged 14 days, (**d2**) 50 μm aged 14 days, (**e1**) 100 μm aged 28 days, (**e2**) 50 μm aged 28 days.

**Figure 7 polymers-16-02593-f007:**
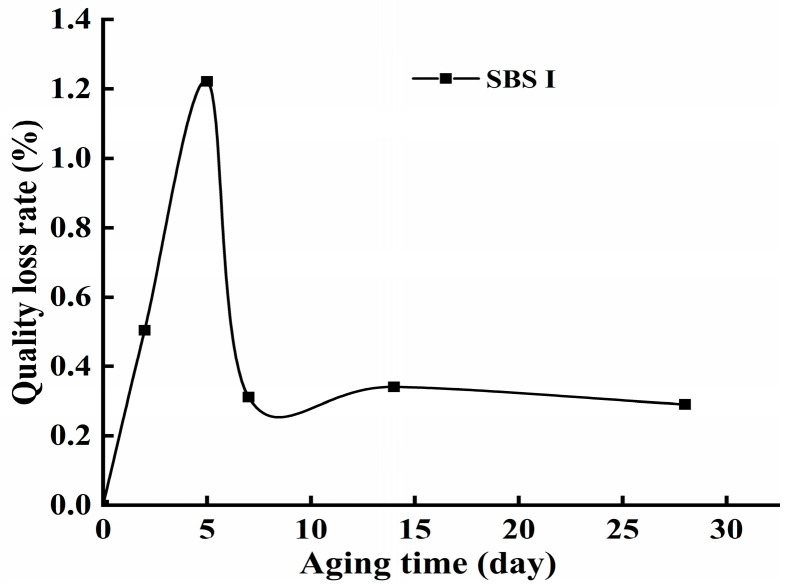
Variation curve of mass loss rate with the aging time.

**Figure 8 polymers-16-02593-f008:**
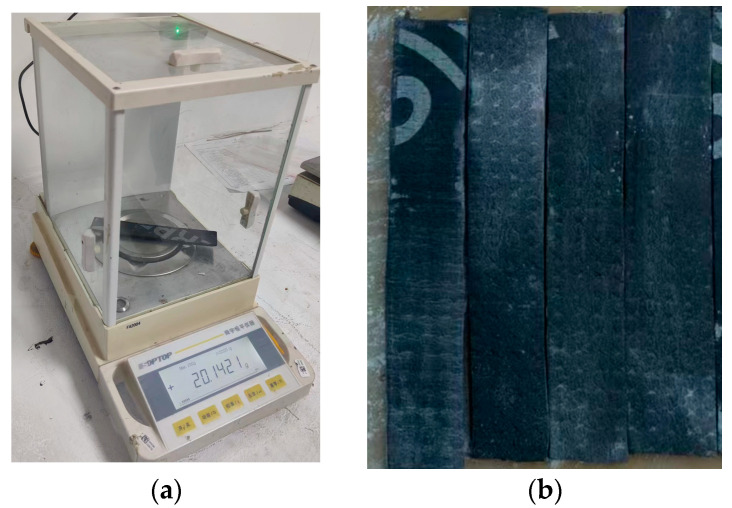
Quality inspection chart: (**a**) electronic scale and (**b**) asphalt membrane with white powder adhesion.

**Figure 9 polymers-16-02593-f009:**
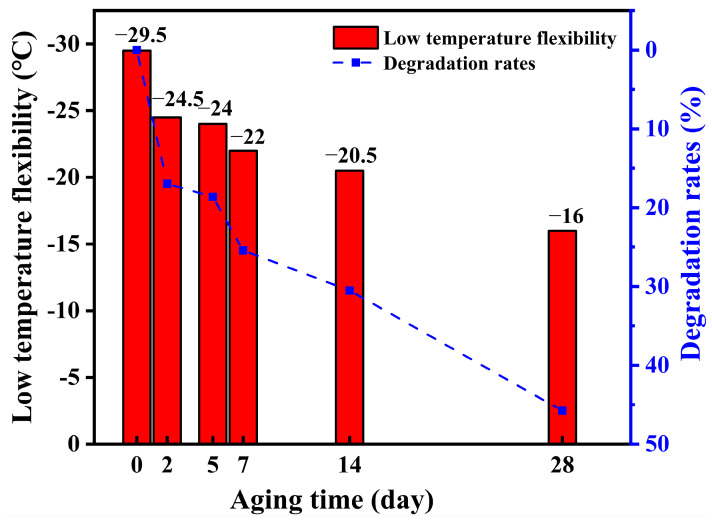
Variation curve of the low-temperature flexibility with the thermal aging time.

**Figure 10 polymers-16-02593-f010:**
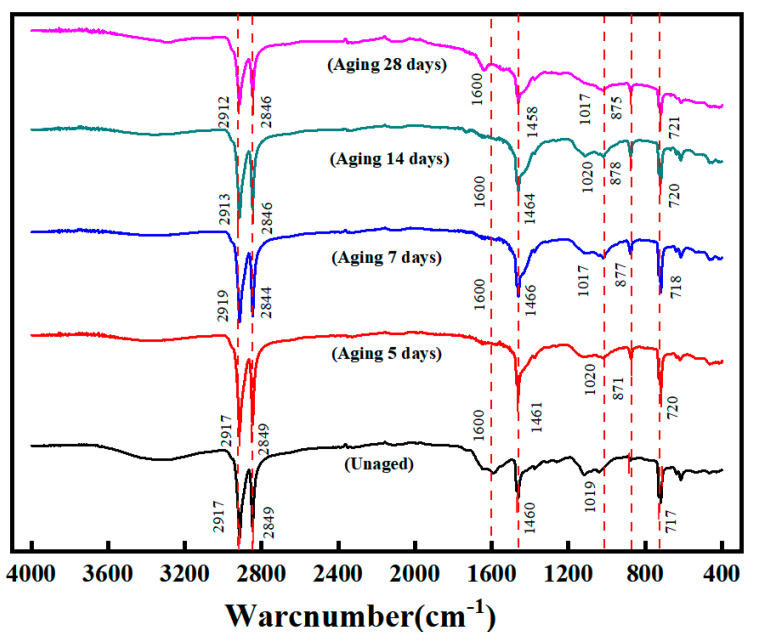
FTIR test graphs after the thermo-oxidation and freeze–thaw cycles.

**Figure 11 polymers-16-02593-f011:**
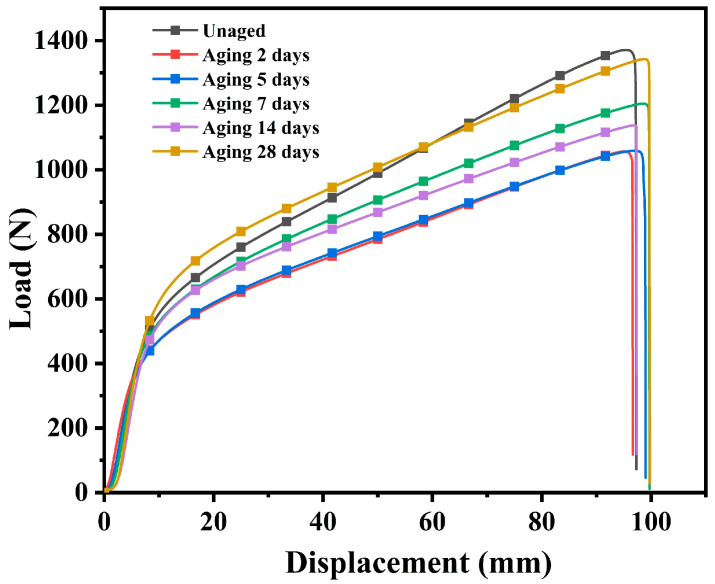
Load-displacement curves of specimens with different aging times.

**Figure 12 polymers-16-02593-f012:**
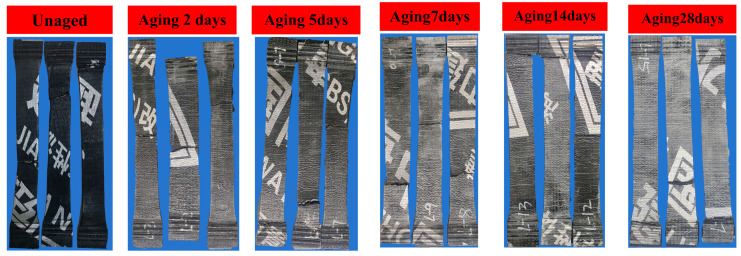
Image of tensile damage pattern.

**Figure 13 polymers-16-02593-f013:**
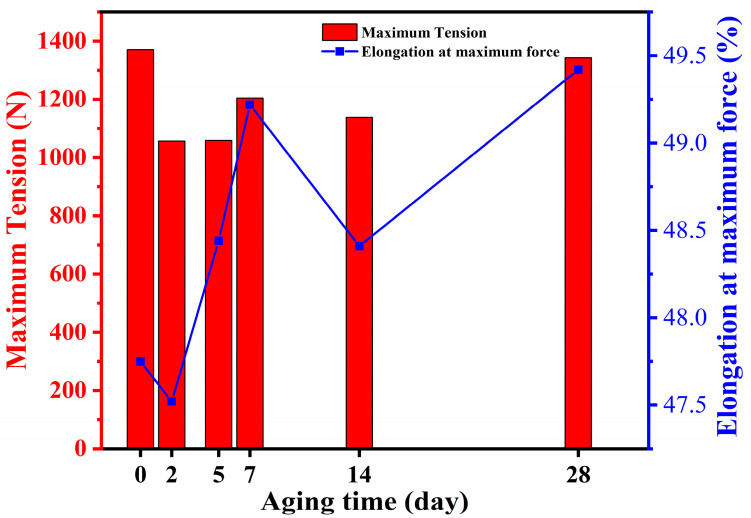
Comparison of tensile strength and elongation of specimens with different aging times.

**Table 1 polymers-16-02593-t001:** Characteristics of the SBS-modified asphalt.

Property	Measured Values
Thicknesses (mm)	3.2
Tensile strength (N/50 mm)	1330/1065
Mass per unit area (kg/m^2^)	3.8
Soluble matter content (g/m^2^)	2310
Maximum applicable temperature	90 °C

**Table 2 polymers-16-02593-t002:** Arrangement of the coupled aging test.

Materials	Aging Temperature (°C)	Freeze–Thaw Cycle Temperature (°C)
SBS-modified asphalt waterproofing membrane	80	−20 to 23
Testing time (day)	2, 5, 7, 14, 28	2, 5, 7, 14, 28

**Table 3 polymers-16-02593-t003:** Peaks observed in the absorption spectra.

Wavenumber (cm^−1^)	Corresponding Peak
2920	Asymmetric stretching vibration of CH2
2850	Symmetric stretching vibration of CH2
1700	Carbonyl (C=O) absorption peak of aldehydes, ketones and acids
1600	Aromatic (C=C) absorption Peak
1030	Sulfoxide group (S=O) stretch vibration absorption
965	Butadiene Double Bond (C=C)
700	Polystyrene bonds

## Data Availability

Data are contained within the article.

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
