# Peer review of "Experimental Investigation on the Performance of SBS-Modified Asphalt Waterproofing Membrane by Thermo-Oxidative Aging and Freeze–Thaw Cycle"

_polymers, 2024, doi:10.3390/polym16182593_

Round 1
Reviewer 1 Report
Comments and Suggestions for Authors
Experimental investigation on the performance of SBS-modified asphalt waterproofing membrane by thermo-oxidative aging and freeze-thaw cycle
In the paper titled “Experimental investigation on the performance of SBS-modified asphalt waterproofing membrane by thermo-oxidative aging and freeze-thaw cycle” the authors consider the extreme weather conditions in determing the performance of awtaer proofing material. Summer and winter conditions were simulated and aging test was conducted.
SBS-modified asphalt waterproofing membranes were subjected to FTIR. Results showed reduction in low temperature flexibility and tensile strength.
The authors shall consider the following points and revise the paper accordingly.
1. The supplier of the mterial and details of the material shall be added. Is the water proof membrane self made or readily available?
2. In 2.2 Test program, the first sentence “To simulate the performance degradation of the waterproofing membrane in high temperature and freeze-thaw environment, SBS modified asphalt waterproofing membrane carried out thermo-oxidative aging and freeze-thaw cycle test, in which the thermo oxidative aging to consider the highest temperature of the roof and external wall in summer [21], set to 80 ℃, and aging time as a variable” shall be made into two or more simpler sentences.
3. It was said that the authors have observed some odour, several folds, appwrance of yellow color and bulge. What is the reason for this kind of behavior.
4. In Page NO. 5, Line No. 181 has to be changed. “diagram” shall be changed as “image”. The sentence “From the microscopic ………… ” shall be changed for clarity”. Authors shall discuss this with reference.
5. In Table 2, Freeze-thaw cycle temperature -20-23C shall be changed as -20 to 23°C.
6. Section 3.2 shall be compared with those in literature.
7. How many specimens were tested for each variant?
8. What is the reason for fluctuation in the Load Vs displacement curve in tensile testing?
9. Figure 12 caption shall be changed . “Diagram” shall be changed with “image”.
10. If possible some more recently published papers shall be cited.
Comments on the Quality of English LanguageSlight English Polishing is needed to improve the paper. Some of them are given in the comments to the author section.
Author Response
Comments 1:The supplier of the mterial and details of the material shall be added. Is the water proof membrane self made or readily available?
Response 1: Thank you for your valuable comment.We have added the details of the material and the supplier information in the revised manuscript. The waterproof membrane used in this study was commercially sourced and not self-made. This information has been clarified in the updated version of the manuscript.
“All test materials were sourced from the same batch produced by Oriental Yuhong Enterprise. The model of the roll is SBS I PY PE PE 3, where: SBS refers to the use of modified bitumen with styrene-butadiene-styrene block copolymer. I denotes that it is a Type I waterproofing roll-roofing. PY stands for the polyester substrate, indicating that the base material of the roll is polyester fabric. PE represents the polyethylene membrane used as the surface coating material. 3 specifies that the thickness of the roll is 3 mm. Thus, SBS I PY PE PE 3 describes a 3 mm thick SBS-modified bitumen waterproofing roll-roofing material with a polyester substrate and a polyethylene surface coating.”
Comments 2:In 2.2 Test program, the first sentence “To simulate the performance degradation of the waterproofing membrane in high temperature and freeze-thaw environment, SBS modified asphalt waterproofing membrane carried out thermo-oxidative aging and freeze-thaw cycle test, in which the thermo oxidative aging to consider the highest temperature of the roof and external wall in summer [21], set to 80 ℃, and aging time as a variable” shall be made into two or more simpler sentences.
Response 2: Thank you for the suggestion. We have revised the first sentence in Section 2.2 to make it clearer by breaking it into two simpler sentences. This change improves readability while maintaining the original meaning.
“To simulate the performance degradation of the waterproofing membrane under high-temperature and freeze-thaw conditions, a thermo-oxidative aging and freeze-thaw cycle test was conducted on the SBS-modified asphalt waterproofing membrane. The thermo-oxidative aging test considered the highest temperatures that roofs and external walls can reach during summer [21], which was set at 80°C, with aging time as a variable.”
Comments 3:It was said that the authors have observed some odour, several folds, appwrance of yellow color and bulge. What is the reason for this kind of behavior.
Response 3: Thank you for your insightful question. The observed behavior, including the odor, wrinkling, yellow discoloration, and surface bulging, is likely due to the oxidative degradation of the asphalt in the waterproofing membrane. This degradation can lead to the release of volatile organic compounds (VOC), causing an odor, while the yellowing and surface changes are likely due to the oxidation of bitumen and the breakdown of polymer additives, leading to brittleness and surface deformation. These observations have been discussed further in the revised manuscript.
“After thermo-oxidative aging, the coil cladding coating gradually hardens, and the rectangular specimen undergoes minor deformation. The volatilization of light components in the asphalt leads to the release of an odor, with the surface emitting a distinct smell of heated asphalt. This is a typical feature of the sub-oxidative degradation of asphalt rolls[23]. The surfaces of specimens with different aging durations are shown in Figure 5. The surface of the unaged specimen was smooth, with a clear texture and glossy appearance, as shown in Figure. a. As the aging time increased, the edges of the specimen surface began to bulge, and the surface turned a yellowish hue. This observation aligned with Zhou et al.'s findings on the broader dispersion of composite modifier chromatographic peaks after aging[24]. A small number of wrinkles appeared on the surface of the modified asphalt after aging, which may have been attributed to the differing thermal expansion coefficients of asphalt and polymers, resulting in uneven contraction of the material surface.”
Comments 4:In Page NO. 5, Line No. 181 has to be changed. “diagram” shall be changed as “image”. The sentence “From the microscopic ………… ” shall be changed for clarity”. Authors shall discuss this with reference.
Response 4:Thank you for your valuable suggestion. We have updated the term 'diagram' to 'image' on Page 5, Line 181, as requested. Additionally,we have revised the discussion of Figure 6 to improve clarity and have included relevant references to support the explanation. Specifically, we have added citations related to the effects of temperature stress and freeze-thaw cycles on the surface damage of asphalt binders. The added references provide further evidence that the observed microscopic surface roughness and micro-cracks are due to the expansion and contraction during freeze-thaw cycles, which exceed the material's stress relaxation limits, leading to damage [25, 26].
“Figure 6 shows the unaged and coupled aging SBS-modified asphalt waterproofing membrane electron microscope scanning image. From the microscopic view, the surface of the unaged specimen is smooth, flat, and dense. As aging progresses, the surface gradually becomes rough and develops a layered, concave-convex structure. This corresponds to the macroscopic observation of small pits and cracks of varying degrees on the surface of the SBS-modified bitumen waterproofing membrane. It is hypothesized that the temperature stress generated during the freeze-thaw cycle causes surface damage to the asphalt binder. Simultaneously, the volume expansion and contraction during the freeze-thaw cycle lead to internal intermolecular interactions. When the temperature stress is insufficient to counteract the material's stress relaxation properties and exceeds the tolerance limit of the asphalt mixture, tiny cracks form [25]. Furthermore, as the number of freeze-thaw cycles increases, the resulting damage becomes more pronounced [26].”
Comments 5: In Table 2, Freeze-thaw cycle temperature -20-23C shall be changed as -20 to 23°C.
Response 5:Thank you for pointing that out. We have revised Table 2 to change the freeze-thaw cycle temperature range from '-20-23' to '-20 to 23' as requested.
Comments 6: Section 3.2 shall be compared with those in literature.
Response 6:Thank you for your valuable comment. We have revised Section 3.2 to include a comparison of our findings with those reported in the literature.
“When measuring the mass, the asphalt coil softened from heat, and some white powder adhered to its surface crevices, leading to a decrease in mass loss. This occurred during the thermal-oxidative aging test of the coil. To prevent the specimen from adhering to the internal platform of the film oven, putty powder was applied. As aging time increased, the specimen's surface became more adhesive, causing the putty powder to stick to it. The post-processing could not completely remove the powder, resulting in an apparent increase in the specimen's mass. Yang et al. also compared the thermal aging and mass loss of Type I SBS-modified asphalt and found that the rate of mass loss was positively correlated with thermal aging time, though the curve did not follow a linear pattern [20]. The aging of asphalt due to oxidation mechanisms results in a reduction in mass [27].”
Comments 7:How many specimens were tested for each variant?
Response 7:Thank you for your question. For the low-temperature flexibility tests, at least 5 specimens were tested at each temperature to ensure accurate determination of the cracking temperature. For tensile tests conducted after thermo-oxidative and freeze-thaw cycle aging, 3 specimens were tested in each group. This information has been clarified and included in the revised manuscript
“For low-temperature flexibility tests, test at least 5 specimens for each coupled aging time, and ensure that at least 4 samples crack in each set of tests. For tensile tests, use 3 specimens per set.”
Comments 8:What is the reason for fluctuation in the Load Vs displacement curve in tensile testing?
Response 8:Thank you for your insightful question. The fluctuation in the maximum tensile load across specimens with different aging times is likely due to a combination of factors, including changes in the internal structure of the material, dynamic interactions between cross-linking and degradation, and the influence of micro-crack propagation. These factors can cause irregular stress distribution, resulting in the non-monotonic behavior observed in the maximum tensile load. We have included a more detailed discussion of these mechanisms in the revised manuscript.
“Moreover, changes in the internal structure of the material during the aging process may have contributed to this behavior. The material likely experienced a dynamic equilibrium between cross-linking and degradation. At certain stages, cross-linking enhanced the material’s strength, whereas at other stages, degradation became dominant, resulting in a reduction in strength. Additionally, the gradual formation and propagation of microcracks may have further contributed to the fluctuations in tensile force[30].
Comments 9:Figure 12 caption shall be changed . “Diagram” shall be changed with “image”.
Response 9: Thank you for your suggestion. We have updated the caption of Figure 12 by changing 'Diagram' to 'Image' as recommended.
Comments 10: If possible some more recently published papers shall be cited.
Response 10:Thank you for your valuable suggestion. We have reviewed the relevant literature and incorporated more recently published papers in the revised manuscript.

Reviewer 2 Report
Comments and Suggestions for Authors
1.Table 1 does not give complete information about the materials tested, e.g. no values for thickness and filler types, moisture absorption, temperature range, etc. The name mentioned in line 126 (SBS I PY PE PE PE 3) shows the filler classification and top and bottom surface finishes, but the abbreviations are not disclosed.
2.The quality of the images in Figure 5, page 6, line 178 does not allow us to assess the difference between samples after aging days 2, 5, and 7.
3.Figure 6 is missing the SEM image of aging on day 2 of the test. the authors should add it.
4.Line 201 mentions a white powder whose composition has not been determined and therefore its contribution to the reduction in quality loss after 14 days is unclear. ABs should clarify.
5.From which literature sources or data obtained on lines 212-218 were conclusions drawn about chemical decompositions and interactions contributing to the deterioration of the flexibility test data.
6.According to the literature review it is not clear for what purpose this work was done, because no new additives were made, but materials already known and currently in use were tested. As it is known, these materials already have known characteristics in terms of operating temperature range.
Author Response
Comments 1: Table 1 does not give complete information about the materials tested, e.g. no values for thickness and filler types, moisture absorption, temperature range, etc. The name mentioned in line 126 (SBS I PY PE PE PE 3) shows the filler classification and top and bottom surface finishes, but the abbreviations are not disclosed.
Response 1: Thank you for your valuable suggestions. We have revised Table 1 to add detailed information about the materials tested, such as thickness and maximum application temperature. In addition, we have explained the abbreviation “SBS I PY PE PE PE PE 3”, detailed the top and bottom surface treatments, and added to the manuscripts.
“All test materials were sourced from the same batch produced by Oriental Yuhong Enterprise. The model of the roll is SBS I PY PE PE 3, where: SBS refers to the use of modified bitumen with styrene-butadiene-styrene block copolymer. I denotes that it is a Type I waterproofing roll-roofing. PY stands for the polyester substrate, indicating that the base material of the roll is polyester fabric. PE represents the polyethylene membrane used as the surface coating material. 3 specifies that the thickness of the roll is 3 mm. Thus, SBS I PY PE PE 3 describes a 3 mm thick SBS-modified bitumen waterproofing roll-roofing material with a polyester substrate and a polyethylene surface coating.”
Comments 2: The quality of the images in Figure 5, page 6, line 178 does not allow us to assess the difference between samples after aging days 2, 5, and 7.
Response 2: Thank you for your suggestions. The images in Figure 5 are scanning electron microscope (SEM) images of specimens with different aging durations, which reflect the macro aging process through microscopic changes, such as the transition from a smooth to a rough surface. These images are intended to show the progression of surface deterioration. We understand that the image clarity is critical, and while these are the clearest images available, we have replaced them with the best possible versions. Additionally, we have further elaborated in the text to better explain the observed changes between aging days 2, 5, and 7.
Comments 3:Figure 6 is missing the SEM image of aging on day 2 of the test. the authors should add it.
Response 3: Thank you for your valuable suggestion. We have now added the missing SEM image for day 2 of the aging test in Figure 6. This addition provides a more complete visual representation of the aging process, allowing for a clearer comparison of the changes in surface morphology over different aging periods
Comments 4:Line 201 mentions a white powder whose composition has not been determined and therefore its contribution to the reduction in quality loss after 14 days is unclear. ABs should clarify.
Response 4:Thank you for your comments. We have clarified the source of the white powder mentioned in line 201 and provided further explanation of its composition. We have revised the manuscript to include this explanation and its possible role in influencing the reduction in mass loss after 14 days.
“From the figure, it can be seen that after different aging times, the quality has decreased, which was because, under heat, the light component in the bitumen will volatilize from the surface of the material, resulting in a decrease in the quality of the waterproofing membrane. The rate of quality loss was faster in the first 5 days and slower in the 7th to 28th day, but the overall trend was positively correlated with the aging time. When measuring the mass, the asphalt coil softened from heat, and some white powder adhered to its surface crevices, leading to a decrease in mass loss. This occurred during the thermal-oxidative aging test of the coil. To prevent the specimen from adhering to the internal platform of the film oven, putty powder was applied. As aging time increased, the specimen's surface became more adhesive, causing the putty powder to stick to it. The post-processing could not completely remove the powder, resulting in an apparent increase in the specimen's mass. Yang et al. also compared the thermal aging and mass loss of Type I SBS-modified asphalt and found that the rate of mass loss was positively correlated with thermal aging time, though the curve did not follow a linear pattern [20]. The aging of asphalt due to oxidation mechanisms results in a reduction in mass [27].”
Comments 5:From which literature sources or data obtained on lines 212-218 were conclusions drawn about chemical decompositions and interactions contributing to the deterioration of the flexibility test data.
Response 5:Thank you for your insightful question. The conclusions drawn in lines 212-218 regarding chemical decomposition and interactions affecting the flexibility test data are based on the following mechanisms: Firstly, during thermo-oxidative aging, the active groups in asphalt break down, forming more reactive free radicals, which accelerate the reaction with oxygen, leading to hardening and brittleness, and a significant decrease in low-temperature flexibility [28]. Secondly, the butadiene segments in SBS also undergo oxidation, forming active free radicals that cause molecular chain scission and degradation, reducing the rubber elasticity and diminishing the improvement in low-temperature performance [29]. These conclusions are supported by the references provided and are now further detailed in the revised manuscript
Comments 6:According to the literature review it is not clear for what purpose this work was done, because no new additives were made, but materials already known and currently in use were tested. As it is known, these materials already have known characteristics in terms of operating temperature range.
Response 6:Thank you for your valuable feedback. While it is true that the materials tested in this study are already known and widely used, the primary purpose of this work was to evaluate the performance of these materials under specific combined aging conditions (thermo-oxidative aging and freeze-thaw cycles). Although the operating temperature range of these materials is known, their long-term durability, mechanical properties, and chemical stability under these combined aging conditions have not been fully explored. The aim of this research is to simulate and analyze how these materials behave in more extreme and realistic conditions to provide further insights into their practical performance in real-world applications
